# Fournier’s Gangrene under Sodium–Glucose Cotransporter-2 Inhibitors Therapy in Gynecological Patients

**DOI:** 10.3390/ijerph19106261

**Published:** 2022-05-21

**Authors:** Adriana Serrano Olave, Ana Isabel Bueno Moral, Carmen Martínez Bañón, Ernesto González Mesa, Jesús S. Jiménez López

**Affiliations:** 1Obstetrics and Gynecology, Materno-Infantil Hospital Regional Universitario Malaga, Avd Arroyo de los Angeles S/N, 29011 Malaga, Spain; anaibuenom@gmail.com (A.I.B.M.); carmenmrtba@gmail.com (C.M.B.); egonzalezmesa@gmail.com (E.G.M.); jesuss.jimenez.sspa@juntadeandalucia.es (J.S.J.L.); 2Surgical Specialties, University of Malaga, 29010 Malaga, Spain

**Keywords:** gangrene, SGLT-2 inhibitor, diabetes, genital lump, gas, emphysema, surgery

## Abstract

Fournier’s gangrene (FG) is a serious pathology of the soft tissues and fascia of the perineum and genital region with a high morbidity and mortality rate. In recent years, the SGLT-2 inhibitor oral antidiabetic has been related to this entity. According to the new warnings from the main drug agencies, a compilation of cases has been initiated to establish or deny a possible causal relationship. Most of these cases have been reported in men. However, it is important not to underestimate this entity in the gynecological field, since it is extremely serious and requires intense and rapid aggressive treatment based on surgery and empiric antibiotherapy. Later, some cares are needed to involve surgical reconstruction of the defects introduced by debridement. As a result of the low incidence of FG, clinical trials’ data may be insufficient to robustly assess this issue because of the limited numbers of participants. Real-world evidence may help to clarify the association between SGLT2i and FG. The aim of this review is to describe and compare the reported cases of GF in diabetic women who received SGLT2 inhibitors as antiglycemic agents.

## 1. Introduction

Fournier’s gangrene was described by Jean Alfred Fournier in 1883. Alfred, a French dermatologist and venereologist, described this acute-onset as a rapidly progressing perineal disease in previously healthy young men [1]. This condition is a rare, life-threatening bacterial necrotizing infection of the perineum. The main risk factors for Fournier’s gangrene are diabetes, obesity, immunosuppression (such as HIV), alcoholism, smoking, male sex, and the use of cytotoxic drugs [2]. Although many of the associated comorbid risk factors are common diseases, FG is rare. The published literature on its incidence in men and women is quite limited. Therefore, analysis from the US Inpatient Database of 593 civilian hospitals in 13 states in 2001 and 21 states in 2004 report that Fournier’s gangrene occurs in 1.6 of every 100,000 males per year, primarily between 50 and 79 years (3.3 of every 100,000). Nevertheless, the small number of Fournier’s gangrene cases in women precludes any meaningful incidence analysis [3,4]. The infection is usually polymicrobial due to an aerobic and an anaerobic bacteria. Traditionally, the diagnosis of this entity was made by clinical examination. Thus, the most common symptoms are scrotal edema, local pain, hyperemia, pruritus, crepitus (between 19% and 64% of cases), fever, and the presence of foul-smelling secretions. Due to a progressive increase in imaging studies, the diagnosis of this entity has become more common, computed tomography (CT) of the abdomen and pelvis being the most appropriate scan. It accurately evaluates the extent of the necrosis, including the possible dissemination to the retroperitoneum [5]. The treatment is aggressive combining surgical debridement and broad-spectrum antibiotics. Reported mortality rates range from 3% to 45%, especially associated with comorbidities such as those previously mentioned [6].

In August 2018, the US Food and Drug Administration (FDA) issued a warning that Sodium glucose co-transporter 2 inhibitors (SGLT2i) may cause FG [7]. The underlying physiopathological mechanisms are not fully clear, but there appears to be greater endothelial damage at the microvascular level. Later, in January 2019, the Spanish Society of Gynecology & Obstetrics (SEGO), the European Medicines Agency (EMA) and the Spanish Agency for Medicines and Health Products (AEMPS) published a statement mentioning the risk of FG in women treated with SGLT 2 inhibitors [8]. SGLT2i are relatively new antihyperglycemic agents that have become popular in the treatment of diabetes due to their favorable cardiac and renal outcomes. The EMA in 2012 and the FDA in 2013 approved these drugs for patients with type 2 diabetes (T2DM) as an adjunct to diet and exercise to improve glycemic control; including canagliflozin, dapagliflozin, empagliflozin, and ertugliflozin [7,9]. The SGLT2 cotransporter is located almost exclusively in the kidney and it is responsible for the reabsorption of 90% filtered glucose by the glomerulus. Thus, its inhibition improves insulin resistance and decreases glycosylated hemoglobin (HbA1c) values. By inhibiting the reabsorption of glucose and sodium from the renal tubule, SGLT2i stimulates glycosuria and natriuresis, reducing blood glucose, body weight and blood pressure [9]. Nevertheless, this mechanism is independent of insulin, and therefore it makes an interesting drug in combined therapies [10]. The most common adverse reactions are mild genital and urinary tract infections [1]. In contrast, there are studies where the risk of serious and non-serious UTI events among patients treated with SGLT-2 inhibitors was similar to that of those treated with other antidiabetics [11]. Additionally, more serious and life-threatening side effects have been found including ketoacidosis, acute kidney injury, increased amputation rates, and Fournier’s gangrene [8,9,10].

This review arises from three cases that occurred in our center, the Materno-Infantil Hospital University Regional of Malaga, between January 2019 and December 2021. The main objective of this review is to describe and compare Fournier’s gangrene in diabetic women who received SGLT2 inhibitors as antiglycemic agents as well as making an analysis of the literature collected so far on the female sex. Most of the cases published to date have been in men. Women also suffer from this pathology that is sometimes underestimated and diagnosed too late. For this reason, through the analysis of clinical data, we want to help the medical practitioners in the early diagnosis of FG, being especially important signs and symptoms as well as complementary tests to support clinical suspicion. The risk factors that make our patients vulnerable will be highlighted. Additionally, of course, the ultimate treatment strategy will be defined to obtain the best possible results.

## 2. Case Presentation

Moreover, we will present three cases that took place in our center in a chronological order. Thus, we will analyze the anamnesis, exploration, complementary tests, some images and treatment, among others.

Case 1: In January 2019, a 48-year-old female at the emergency department presented an increased swelling in relation to a painful lump of 5 days’ duration, more exacerbated in the last 48 h. This pain radiated to the area of the deep right iliac fossa and was associated with a fever of 39 °C. The patient had been taking clavulanate amoxicillin for 3 days without clear improvement. Personal history included obesity, dyslipidemia and poorly controlled T2DM for more than 15 years currently on treatment with oral antidiabetics such as Metformin 850 mg-Dapagliflozin 5 mg every 12 h for 19 months with a 9% of HbA1c. She was smoking 15 cigarettes per day. No alcohol or other drugs. She had no previous surgeries.

According to the initial evaluation, the patient was febrile (39 °C), tachycardic (123 beats/min), with blood pressure of 146/78 mmHg and respiratory rate of 22 breaths/min. The examination revealed a huge phlegmonous abscess on the right labia majora that extended to the mons pubis (Figure 1A). Rest of the examination was unremarkable. Laboratory workup revealed blood glucose of 243 mg/dL, mild leukocytosis (16,450/L) with 80% neutrophils and C-reactive protein (RCP) of 492 mg/dL. Normal coagulation times. No more unaltered parameters. A CT of the abdomen and pelvis revealed important inflammatory changes and air inside the soft tissues of the genital area extending to the right iliac and paraumbilical fossa compatible with FG (Figure 1B).

Broad-spectrum intravenous antibiotic treatment with Cefoxitin, Metronidazole and Gentamicin was administered. Tissue culture showed a combination of *Staphylococcus auricularis* and *Bacteroides fragilis* (both anaerobic pathogens). Blood culture was negative. Finally, surgical debridement was performed until the rectus abdominis muscles were completely exposed due to the advanced extent of the disease (Figure 2A). Subsequently, she was transferred to the critical care unit, where after more than four reinterventions (Figure 2B) and supportive measures, she was discharged two months later.

Case 2: In September 2019, an 84-year-old woman came to the emergency room due to a painful genital abscess from three days’ duration. She also had nausea without vomiting. She had non-thermometered dysthermic sensation. She had been receiving treatment with oral Augmentin since the previous day. Medical history included: ex-smoker (>15 cigarettes/day), poorly controlled chronic hypertension despite four drugs, T2DM (30 years complicated evolution with diabetic microangiopathy and diabetic foot) with 8% of HbA1c. No alcohol or other drugs. In addition, her surgical history includes: hysterectomy, appendectomy, hip prosthesis and inguinal hernioplasty. At this moment, she was under antidiabetic treatment with Metformin 1 g/Canagliflozin 50 mg every 12 h for 3 months.

On the initial scan, her general condition was fair. She had blood pressure around 122/52 mmHg, HR 110 bpm and 36 °C of temperature. Right labia majora was enlarged, indurated and erythematous. Additionally, she had a painful ulcer in the lower third. Blood tests revealed hemoglobin 8.9 g/dl, platelets 426,000/L, leukocytosis 37,640/L with 90% neutrophils and RCP of 285 mg/dl. Glycemia at 465 mg/dl. No more unaltered parameters were associated. A CT of the abdomen and pelvis showed subcutaneous edema and air in the perineal area and right vulvar area extending into the presacral soft tissues suggestive of FG (Figure 3).

Despite an immediate intensive surgery, a broad-spectrum antibiotic therapy, and partly due to the patient’s multiple pathologies, she died after 48 h by a multi organ failure. Nevertheless, the patient was in a frail state prior to the infection. Tissue culture was positive for *Streptococcus anginosus* and *Prevotella bivia*. Blood culture was negative. 

Case 3: In January 2021, a 68-year-old morbidly obese smoker woman with chronic hypertension and poorly controlled T2DM (>15 years) arrived to the emergency department with a three-days history presenting a genital lump, hypogastric pain and fever (38 °C). No alcohol or other drugs. Her usual treatment included Linagliptin 5 mg every 24 h for a year and Metformin 1 g-Empagliflozin 5 mg every 12 h for 39 months (3.25 years) with 7.2% of HbA1c. According to the surgical history, doctors highlighted: radical nephrectomy, hysterectomy and double adnexectomy, hernioplasty. Moreover, they made a special mention of the fact that she had a single kidney.

During the examination, she was in good general condition: normal blood pressure, only the temperature of 38 °C stood out. Blood tests showed slight anemia (hemoglobin 10.8 g/dL), normal platelets, marked leukocytosis 24,280/L with 92.8% neutrophilia. Normal coagulation times. Glomerular filtration decreased by 28%, creatinine 1.83 mg/dl (in accordance with her monorenal status). CRP 220 mg/dl. A CT showed soft tissue emphysema extending from the left labia majora to the mons Venus and left inguinal region. They were seen some thickening and trabeculation of fatty tissues around perineum and left femoral region. Radiological findings compatible with FG (Figure 4).

After diagnosis, broad-spectrum antibiotic therapy with Piperacillin/Tazobactam started. Meropenem was subsequently added. She was also urgently operated on for tissue debridement. Up to three reinterventions were necessary (Figure 5). Tissue culture was positive for *Aerococcus urinae* and *Candida albicans*. The blood culture was negative. However, despite efforts and support measures, the patient did not survive and died 14 days later.

## 3. Discussion

Since the issuance of the health alert by the FDA in 2018 and later by the SEGO, EMA and AEMPS in 2019, FG cases under SGLT2 Inhibitors therapy have increased [12,13]. There is a bias called notoriety bias. It is a selection of bias in which a case has a higher probability of being reported if the subject is exposed to the factor studied, which is known or thought to cause the event of interest [14]. Therefore, it could partly explain the increased FG reporting numbers after the warning of different drugs agencies. Although, most cases have been suffered by men, from the Gynecology and Obstetrics service of the Malaga Regional Hospital. We would like to reflect the non-negligible number of female cases. One of the reasons for the relatively lower appearance of FG in women could be related to the better drainage of the perineal region by vaginal secretions. Another reason could be confusing in the initial diagnosis regarding other infections of the genital area.

In the following table, we have represented the three cases described previously and also compare their main characteristics (Table 1).

Among patients with remarkable characteristics, we believe that advanced age possibly exerts a negative influence on prognosis. Regarding medical history, in all cases there was obesity, long poorly controlled type 2 diabetes mellitus and tobacco (smoker or ex-smoker). Median female body mass index (BMI) was 38, which implies that the comorbidities of the patients reported must be considered [10]. In other published articles, it was described that the average age of presentation of FG is 50–60 years. Among other risk factors are, as in our cases, diabetes mellitus, obesity and alcoholism [15,16,17]. In addition, it seems that the female sex is an added risk factor for mortality [16] as well as tobacco and obesity seem to play a role in enhancing the adverse effects of diabetes and the risk of FG. All patients shared a poorly adjusted DM2 at the time of diagnosis with blood glucose levels above 180 mg/dl [18]. Hb1Ac analysis could refine the diabetes in a severe scale. Our patients all presented high values (9%, 8% and 7.2%, respectively), which may reflect an incomplete diabetic control. The measurement of glycated Hb is a laboratory test widely used in diabetes to know if the patient’s control over the disease has been good during the last three or four months (although there are doctors who only consider the last two months). Surgical history does not seem relevant. 

Regarding treatment, all patients were taking a compound of Metformin and SGTL-2 inhibitor (Dapagliflozin, Canagliflozin and Empagliflozin). Another noteworthy fact is that no case of Fournier’s gangrene involved ertugliflozin, which could be due to its shorter time to market. The mean was about 20 months, although more cases are needed to estimate a statistically significant minimum treatment time. The underlying mechanism is unknown. Elevated blood glucose levels (above 180 mg/dl) and additional SGLT2 therapy can lead to a state of glycosuria, favoring urinary tract infections. This is associated with local immunodeficiency and deficient microvascularity (T2DM and other comorbidities) which may promote FG in certain patients [19]. 

Three patients went to the Emergency Department due to a painful genital lump. In addition, fever was present in all cases. The most common anatomic region of gangrene involvement was the labia followed by perineum and gluteus/buttocks. Blood tests were unremarkable except for leukocytosis with neutrophilia, which were elevated in CRP. In any case, coagulation or hepatorenal function was altered. In general, CT allowed the confirmation of the clinical suspicion in our three patients, revealing gas in the soft tissues. Thus, the characteristic image showed air inside tissues, emphysema and subcutaneous edema, in the most severe cases passing through the muscle and bordering on large vessels. However, it is often enough for diagnosis. Usually, the most frequent isolated aerobe is Escherichia coli and the anaerobe Bacteroides fragilis [18]. Nevertheless, in our review, B. fragilis was only related to one case and E. coli was not found. In 6 out of 10 cases, the infection is polymicrobial, which means the use of several antibiotics initially, with subsequent modification of the schedule based on culture findings [19]. All blood cultures were negative. 

Truthfully, the initial treatment algorithm in our three patients was quickly applied through empirical antibiotic therapy and surgical debridement. There is no consensus in the literature on the optimum antimicrobial treatment for FG. Our three patients, as the Infectious Diseases Society of America (DSA) and the Spanish Society of Infectious Diseases and Clinical Microbiology (SEIMC) recommend, received a broad-spectrum treatment based on the use of a Carbapenem, a Cephalosporin or a beta-lactam/beta-lactamase inhibitor (Meropenem in all cases) plus Vancomycin or Daptomycin for methicillin-resistant Staphylococcus aureus (MRSA) coverage, and Clindamycin for its antitoxin activity against Streptococci or Staphylococci [18]. Subsequently, antibiotics were regulated according to antibiograms. Outcome was truly negative, with only one survivor. It is unclear whether patients could be started on SGLT2 inhibitors again after complete remission, although we prefer not to restart it [11]. Likewise, in all cases, they needed several surgeries, most of them up to four interventions. The length average of staying was two months. The oldest patient died 48 h after the surgery. Despite all the efforts and the rapid response, the outcome was fatal in two of the patients and the survivor presented high comorbidity. Answering the question of whether the female gender is a risk factor for mortality in patients with FG, it is associated with a higher incidence of inflammation of the retroperitoneal space and the abdominal cavity. The differences between the male and female genital anatomy could be the reason for the rapid spread of the infection to the retroperitoneum and the fatal outcome in women [19,20,21], although more studies are needed to back it up. 

## 4. Conclusions

Based on the recent warnings from the FDA, SEGO, EMA and AEMPS and the drastic growing popularity of therapy with SGLT2 inhibitors, especially rising quickly worldwide along with the increased needs of diabetic patients with heart disease and obesity, it is important to consider the possible and fatal adverse effects [6,12]. Likewise, genital infections in patients with risk factors such as aforementioned T2DM and tobacco should alert the medical community to rule out FG [12]. Furthermore, it is suggestive to request HbA1c at diagnosis in order to estimate the risk of glycosuria due to SGLT2i and poor diabetic control with FG. It is necessary realizing more scientific analysis between the onset time of FG associated with SGTL-2 inhibitors, as well as studies to generate evidence for a causal connection or improve treatment algorithms for patients with FG [9]. This class of drug has only been on the market for the last nine years, hence the information of its true risk and side effects is limited. Finally, we encourage doctors to voluntarily report all adverse drug effects in order to conduct post-marketing studies to determine the true risk of SGLT2i in diary clinical practice [8].

## Figures and Tables

**Figure 1 ijerph-19-06261-f001:**
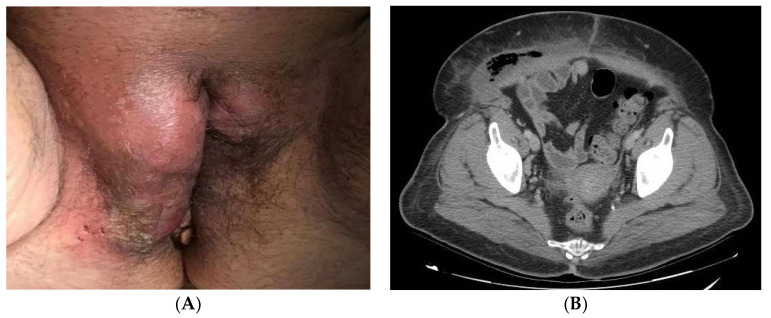
(**A**): Perineal abscess with induration of surrounding tissue. The most common onset site of FG is the labia majora. (**B**): CT of the abdomen and pelvis showing areas with gas revealing necrotic tissue from gangrene.

**Figure 2 ijerph-19-06261-f002:**
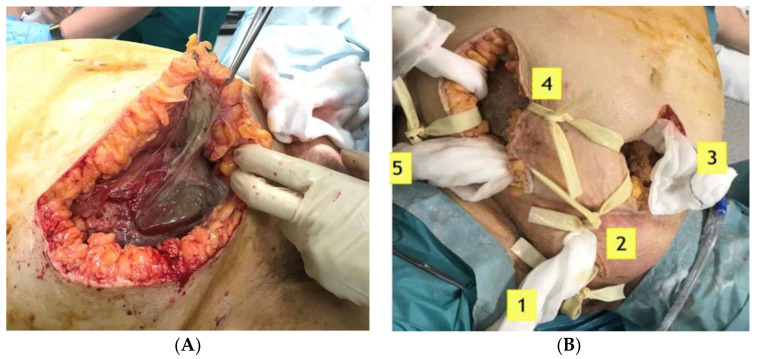
(**A**): Surgical treatment by aggressive debridement, pointing out the necrotic tissue below. (**B**): Reinterventions for removal of dead tissue, vacuum-assisted closure therapies, healthy tissue grafts are common in FG.

**Figure 3 ijerph-19-06261-f003:**
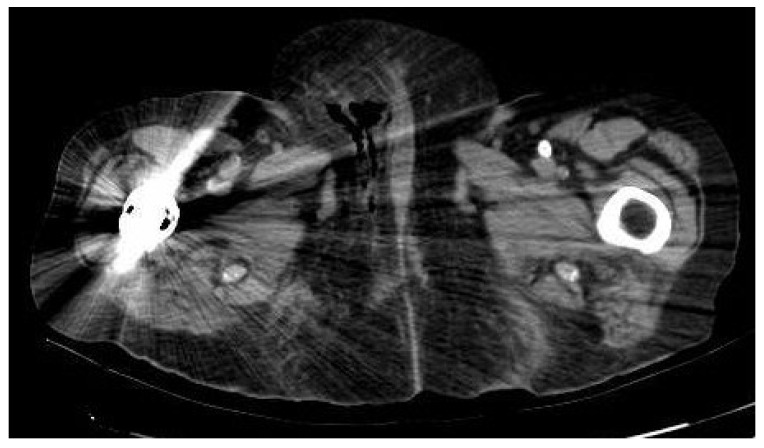
CT of the abdomen and pelvis showed subcutaneous edema and air inside the soft tissues of the genital area extending into the presacral soft tissues of FG. The hip replacement slightly distorts the image.

**Figure 4 ijerph-19-06261-f004:**
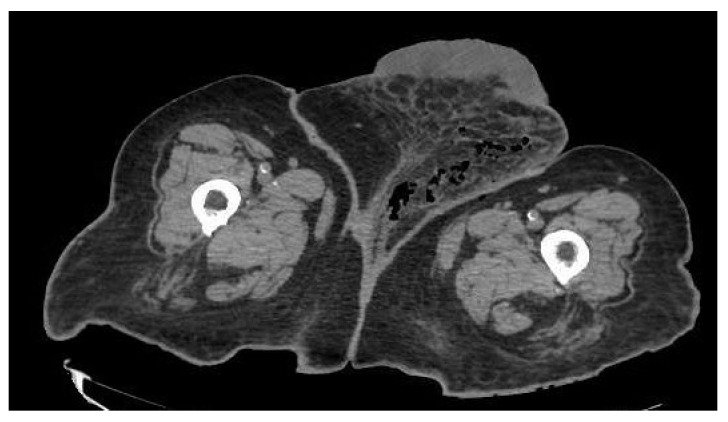
CT of the abdomen and pelvis showed a FG which involves a soft tissue emphysema extending from the left labia majora to left femoral region, associated to a thickening and trabeculation of fatty tissues around them.

**Figure 5 ijerph-19-06261-f005:**
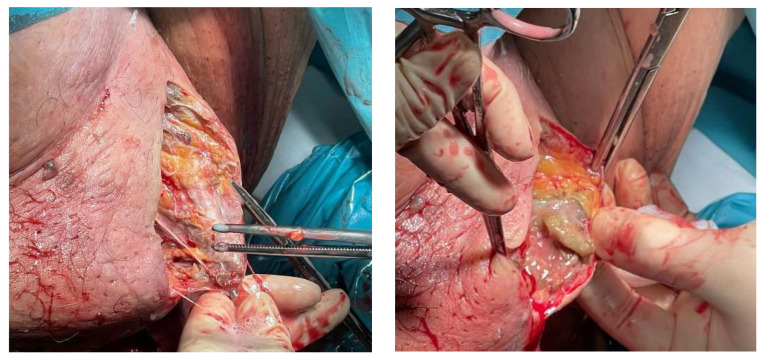
Necrotic tissues exposed during surgery prepared for debridement. The surgical technique involves removing as much death tissue as possible, leaving open wound to oxygenate it.

**Table 1 ijerph-19-06261-t001:** Comparison table. Representation of the three cases and their main characteristics.

	Case 1	Case 2	Case 3
**Age**	48	84	68
**Medical history**	obesity, poorly controlled T2DM (>15 years) and tobacco		
**Surgical history**	No	Yes	
**Current treatment**	Metformin 850 mg-Dapagliflozin 5 mg every 12 h for 19 months	Metformin 1 g/Canagliflozin 50 mg every 12 h for 3 months	Metformin 1 g-Empagliflozin 5 mg every 12 h for 39 months
**HbA1c level**	9%	8%	7.2%
**Signs and symptoms**	painful genital lump and/or fever > 38 °C		
**Blood analysis**	leukocytosis with neutrophilia and CRP elevation		
**Abdominal CT**	soft tissue emphysema, subcutaneous edema and areas with gas revealing necrotic tissue from gangrene		
**Tissue culture**	*Staphylococcus auricularis* and *Bacteroides fragilis*	*Streptococcus anginosus* and *Prevotella bivia*	*Aerococcus urinae* and *Candida albicans*
**Blood culture**	negative		
**Treatment**	empirical antibiotic therapy and surgical debridement		
**Outcome**	cured	deceased	Deceased
**Antibiotic therapy**	Meropenem + Daptomycin, later Piperacillin/Tazobactam + Clindamycin	Meropenem + Vancomycin + Clindamycin	Meropenem + Daptomycin + Clindamycin
**Timing**	25 days	5 days	16 days
**Isolate with residence**	Yes	Yes	Yes
**Surveillance swabs**		Yes	
**Duration of the stay**	2 months	5 days	16 days

## Data Availability

All data generated or analyzed during this study are included in this published article.

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
