# Peer review of "Fournier’s Gangrene under Sodium–Glucose Cotransporter-2 Inhibitors Therapy in Gynecological Patients"

_ijerph, 2022, doi:10.3390/ijerph19106261_

Round 1
Reviewer 1 Report
Dear Editor
The article by Adriana Serrano Olave entitled” Fournier’s Gangrene under Sodium–Glucose Cotransporter-2 2 Inhibitors Therapy in Gynecological Patients”
This review Is focus on Fournier's gangrene, a serious pathology of the soft tissues and fascia of the perineum and genital region with a high morbidity and mortality rate, The Authors describe and compare the reported cases of Fournier's gangrene in diabetic women who received SGLT2 inhibitors as antiglycemic agents.
The document opens a very interesting scenario in this setting.
I believe that the document is valid for publication, but need minor revision.
-Please add in table 1 all anbiotic treatment and timing, the presence of isolate with residence, specify if surveillance swabs had been carried out, the duration of the stay
Author Response
Thank you for your suggestions. We have added these data to the original table, which we hope will serve to describe the stay at the hospital in these patients.
Reviewer 2 Report
General comment)
This is a case series to describe three female Fournier’s Gangrene (FG) under SGLT2 inhibitor use. Fournier’s Gangrene is a rare but critical necrotizing fasciitis of the perineal and genital area. The incidence of FG in females is less than in males but not negligible.
The introduction and discussion were not straightforward throughout the manuscript, and it was hard for me to follow what was a novel point. The clinical course of the three patients was comparable with the usual FG in females. Extensive revision of the introduction and discussion seems to be necessary, although the case presentation looks good.
Specific point)
1. (L72-77) Many questions were proposed but remain unanswered. Most of them were difficult to answer from this case series, thus inappropriate. Instead, authors should describe what is novel points in this case series and how these points contribute to the area of female FG.
2. In the case presentation, the HbA1c level should be described, as the authors stated its importance in the conclusion.
3. Characteristics of female FG have been reported elsewhere (PMID 32531467, 19812875, 19139915). Authors need to cite these studies and discuss what are new findings compared to these studies.
4. The possible mechanism why SGLT2 inhibitor is associated with FG should be discussed more, even though no clear evidence exists. It is a big concern that SGLT2 inhibitors may increase UTIs, but a recent report (PMID 31357213) showed the opposite.
5. (L190-191) "Since the issuance of the health alert by the FDA in 2018 and later by the SEGO, EMA and AEMPS in 2019, the number of reported FG cases has increased." This sentence should be supported by evidence. At least, the number of reports about FG related to SGLT2 inhibitor was the highest in 2019 and decreased by years.
6. The conclusion should be much shorter and supported by the findings from the cases.
7. (L270-272) All of the cases presented with perineal pain and origin seemed apparent.
8. (L2730274) This statement should be modest because any risk comparison was not performed between SGLT2 inhibitor user and control in this study.
Minor points)
(L70) "GFR" abbreviation should be spelled out at first.
Table 1, Case 2 was not febrile at presentation.
Ref#8 didn't contain sufficient information. is it a webpage?
Author Response
REVIEWER 2:
- Thank you very much for your suggestion. After it we have withdrawn the questions and made statements about the points to be clarified throughout the article emphasizing their importance.
- We have carried out a search of the HbA1c within the data registered in the Health Center of each patient. Regarding HbA1c, the three patients presented high values (9%, 8% and 7.2% respectively), which may reflects an incomplete diabetic control. Therefore we have added these data in the description of each case as well as in the table and discussion.
- In other published articles (15,16,17) it is described that the average age of presentation of FG is 50-60 years. Among other risk factors are, as in our cases, diabetes mellitus and obesity. In addition, it seems that the FG in the female sex is an added risk factor for mortality. We have added this information in the discussion.
- We have added some sentence to the discussion of said argument. The recent report you have named has been referenced in the article as it counters the previous benchmark studies.
- It is true that since the publication in 2018 of the health alert by the FDA, the cases of FG associated with SGLT2 inhibitors published have increased, as in our paper that are described three cases. To reflect this fact we have added two references: 12 and 13.
- Thank you for your suggestions. We have slightly shortened the conclusion, in addition to be more consistent with the information obtained from the cases.
- We have made changes to this sentence and we hope allow its better understanding.
- This sentence has been deleted as you menciones it was not supported by a case-control study.
Round 2
Reviewer 2 Report
The authors provided appropriate revisions.
Minor point)
P5L199
"FG cases have increased" -> "FG cases under SGLT2 Inhibitors therapy have increased"